# Effectiveness of early versus delayed rehabilitation following rotator cuff repair: Systematic review and meta-analyses

Bruno Mazuquin[1]*, Maria Moffatt[1], Peter Gill[1,2], James Selfe[1], Jonathan Rees[3], Steve Drew[4], Chris Littlewood[1]

1 Department of Health Professions, Faculty of Health, Psychology and Social Care, Manchester Metropolitan University, Manchester, United Kingdom, 2 Northern Care Alliance NHS Group, Manchester, United Kingdom, 3 Nuffield Department of Orthopaedics, Rheumatology and Musculoskeletal Science, University of Oxford and NIHR Oxford Biomedical Research Centre, Oxford, United Kingdom, 4 University Hospitals Coventry and Warwickshire, Coventry, United Kingdom

* b.mazuquin@mmu.ac.uk

## Abstract

### Objective

To investigate the effectiveness of early rehabilitation compared with delayed/standard rehabilitation after rotator cuff repair for pain, function, range of movement, strength, and repair integrity.

### Design

Systematic review and meta-analyses.

### Methods

We searched databases and included randomised controlled trials (RCTs) comparing early with delayed/standard rehabilitation for patients undergoing rotator cuff repair surgery. We assessed risk of bias of the RCTs using the Cochrane RoB 2 tool.

### Results

Twenty RCTs, with 1841 patients, were included. The majority of the RCTs were of high or unclear risk of overall bias. We found substantial variations in the rehabilitation programmes, time in the sling and timing of exercise progression. We found no statistically significant differences for pain and function at any follow-up except for the outcome measure Single Assessment Numeric Evaluation at six months (MD: 6.54; 95%CI: 2.24–10.84) in favour of early rehabilitation. We found statistically significant differences in favour of early rehabilitation for shoulder flexion at six weeks (MD: 7.36; 95%CI: 2.66–12.06), three (MD: 8.45; 95%CI: 3.43–13.47) and six months (MD: 3.57; 95%CI: 0.81–6.32) and one year (MD: 1.42; 95%CI: 0.21–2.64) and similar differences for other planes of movement. In terms of repair integrity, early mobilisation does not seem to increase the risk of re-tears (OR:1.05; 95%CI: 0.64–1.75).

**Data Availability Statement:** All relevant data are within the manuscript and its Supporting information files.

**Funding:** The authors received no specific funding for this work.

**Competing interests:** The authors have declared that no competing interests exist.

## Discussion

Current approaches to early mobilisation, based largely on early introduction of passive movement, did not demonstrate significant differences in most clinical outcomes, although we found statistically significant differences in favour of early rehabilitation for range of movement. Importantly, there were no differences in repair integrity between early and delayed/standard rehabilitation. Most rehabilitation programmes did not consider early active movement as soon as the patient feels able. With ongoing variation in rehabilitation protocols there remains a need for large high quality RCT to inform the optimal approach to rehabilitation after rotator cuff repair surgery.

## Introduction

Shoulder pain is experienced by one in four people at any one time [1] and is one of the most common musculoskeletal pain presentations [2]. Tears of the rotator cuff, the muscles and tendons that closely surround the shoulder joint, are a common cause of shoulder pain [1]. Many people with symptomatic rotator cuff tears can be successfully managed non-surgically; where this treatment proves insufficient, surgery to repair the torn rotator cuff might be offered [3]. In 2018/2019, almost 9,000 surgical repairs were undertaken in the UK National Health Service [4]. However, as the number of operations to repair the rotator cuff increases and surgical techniques advance, there are ongoing uncertainties about the optimal approach to postoperative rehabilitation, a key component of the recovery process [5].

A recent international survey of practice reported that postoperative rehabilitation has not evolved over the last two decades [6]. A generally cautious approach to standard postoperative rehabilitation still remains, and includes sling immobilisation for several weeks [7]. In 2018 a systematic review [8] was published that reported conflicting evidence in relation to early versus delayed/standard rehabilitation following rotator cuff repair. The meta-analyses suggested that early mobilisation did not provide additional clinical benefit in terms of pain or function, although recovery of range of movement of movement was more rapid. It also reported no statistically significant difference in repair integrity, i.e. the number of re-tear events was similar between groups, which is one of the historical concerns for this surgery and the justification for cautious approaches to postoperative rehabilitation. However, most of the randomised controlled trials (RCTs) included in the review were rated as presenting a high risk of bias and data for large tears were lacking.

A number of new RCTs (eight) have been published recently providing the opportunity for up to date evidence. Our aim was to investigate the effectiveness of early compared to standard or delayed rehabilitation following rotator cuff repair for pain, function, range of movement, strength and repair integrity.

## Methods

We reported the systematic review according to the Preferred Reporting Items for Systematic Review and Meta-Analyses (PRISMA) [9]. The protocol was registered in the International prospective register of systematic reviews database (PROSPERO)—PROSPERO 2020 CRD42020209330.

### Eligibility criteria

We included studies that met the following criteria:

- Design: RCT

- Participants: patients aged 18 years or older who had undergone surgical repair of the rotator cuff

- Intervention and comparison: early rehabilitation compared with delayed rehabilitation (as per study definitions).

- Outcomes: pain, function, range of movement, strength and repair integrity.

## Search

We searched MEDLINE, CINAHL, Scopus, SPORTDiscus, SciELO and the Cochrane Library for relevant papers from inception to December 2020. We decided not to limit the searches by date of publication to identify other RCTs that could be relevant but were not included in the previous version. The electronic search strategies were supplemented by hand searching the reference lists and citations of the included RCTs. There was no restriction to date or language of publication. For the search strategies, we combined MeSH terms and keywords such as: rotator cuff, shoulder, shoulder joint, rehabilitation, physiotherapy, immobilisation and RCT. The detailed search strategy is available in the S1 File.

## Screening

Searches results were imported to Rayyan (https://rayyan.qcri.org) where removal of duplicates and screening for eligibility was undertaken by two independent authors (BM and MM). Studies were first screened by their titles and abstracts; the full text was retrieved if further information was needed for a decision.

## Data extraction

Data was extracted by one author (BM) and reviewed by a second author (PG) using a pre-established Excel template. Data extraction included author names, year of publication, country, participants characteristics, characteristics of the intervention and comparator, tools used for outcomes assessment, results for the variables of interest and public and patient involvement and engagement activities. In case of missing data, we contacted the authors via email to request additional available data.

## Risk of bias and grading of evidence

The risk of bias of the included RCTs was assessed by one author (BM) and reviewed by a second author (MM) using the Cochrane risk of bias tool (RoB2) [10]. The Grades of Recommendation, Assessment, Development and Evaluation (GRADE) framework and GRADEpro GDT [11] software were used to rate the certainty of the effect. Outcomes were rated and downgraded according to the presence or absence of factors (risk of bias, inconsistency, indirectness, imprecision) affecting the quality of the body of RCTs included in each outcome.

## Measurement of treatments effect

We generated meta-analyses and forest plots using RevMan 5 (The Nordic Cochrane Centre, Copenhagen, Denmark) [12]. The meta-analyses were presented according to the different outcome measures used across RCTs, for example American Shoulder and Elbow Score (ASES), and follow-up timing. Continuous data was expressed as mean difference and 95%

confidence interval. For dichotomous outcomes, the odds ratio (OR) was used with 95% confidence interval. As re-tear or failure to heal, indicative of repair integrity, is regarded as an unfavourable outcome, we also calculated the Absolute Risk Increase and Number Needed to Harm (NNH) for the repair integrity data. Statistical heterogeneity was tested using the $I^2$ test. In addition, we observed variation in the rehabilitation protocols for exercise dosage and time in the sling; therefore, we used the random effects methods for the meta-analyses. Funnel plots were not created to check for heterogeneity and bias as this is not recommended where meta-analyses have fewer than 10 RCTs, none of our meta-analyses had more than 10 RCTs [13, 14].

## Results

### Study selection

Initially 2142 records were found, after removing duplicates 1805 studies were screened. Of these, 22 were selected for full text review. Another two studies were excluded due to wrong interventions: Baumgarten, Osborn [15] investigated the use of pulley exercises at six weeks and Michael, König [16] explored the effect of continuous passive movement and physiotherapy versus physiotherapy alone. Twenty RCTs were eligible for inclusion; in total, 13 RCTs were included in the meta-analyses (Fig 1).

### Study characteristics

Participants mean age ranged from 54 to 65.4 years. Sample sizes varied from 18 to 206 participants; only nine out of the 20 RCTs (45%) reported a sample size calculation and the majority,

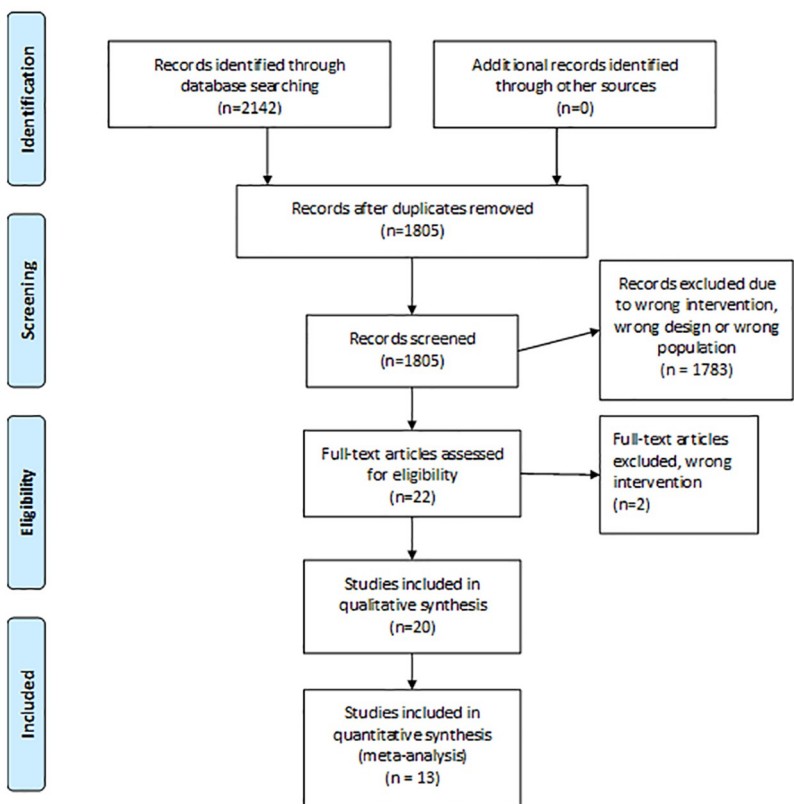

**Fig 1. PRISMA diagram for studies selection.**

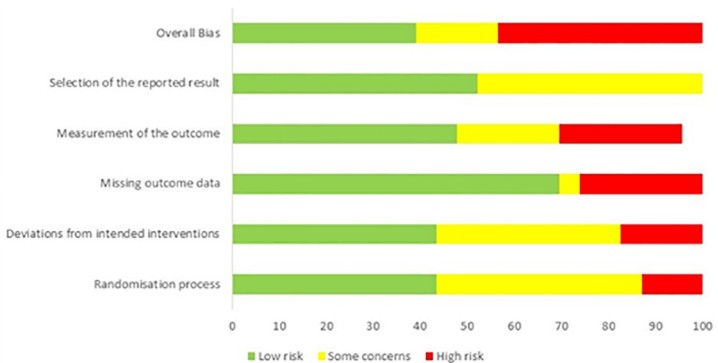

**Fig 2. Risk of bias summary.**

16/20 (80%), recruited patients from a single hospital. The total number of patients from the 20 RCTs was 1841. Only eight of 20 RCTs included patients with large tears [3, 17–23]. There was variation in the use of single or double row repairs as well as additional surgical procedures such as subacromial decompression, acromioclavicular joint excision and biceps tenodesis. Further details are available in the S2 File.

## Risk of bias

Fig 2 summarises the results of the risk of bias assessment. S3 File reports the risk of bias assessment of each RCT. Most RCTs were rated as of unclear or high risk of bias for randomisation process and overall bias.

## Rehabilitation protocols

The method of immobilisation after rotator cuff repair was variable. The majority of the RCTs [3, 20, 21, 24–31](11/20, 55%) reported the use of a standard sling, six RCTs [19, 22, 23, 32–34] used a sling with an abduction component and one RCT [35] used a standard sling for the early group and a sling with an abduction component for the delayed group. The time in the sling was also variable; eight RCTs (40%) [19, 22, 24–27, 30, 33] used a six-week period for both groups. Three RCTs prescribed a sling for the early group to be used for comfort only and discarded as the patient felt appropriate [3, 20, 21]. One RCT did not prescribe a sling for those in the early group [31].

Exercise progression was similar across RCTs, starting with passive exercises, moving to active-assisted, active and then resisted exercises. However, there was variation in the timing for the exercise progression and information about frequency and intensity of exercises. All studies had a time driven protocol at all stages except Littlewood, Bateman [3], Sheps, Bouliane [20], Sheps, Silveira [21]. Littlewood, Bateman [3] used a patient-led approach using acceptable symptom response to prescribe and progress exercises, regardless of the postoperative time. Sheps, Bouliane [20] used a similar protocol to Littlewood, Bateman [3]; however, there is limited information about the protocol after six weeks postoperative. Sheps, Silveira [21] also used a patient-led approach; only pain-free active movements were permitted but exercise progression was based on the number of weeks postoperative. More details about the rehabilitation programmes are available in the S4 File.

**Table 1. Meta-analyses of pain at rest by Visual Analogue Scale (a negative effect estimate favours early rehabilitation).**

| Outcome measure | Follow-up | Number of studies | Total sample size | Effect estimate MD [95% CI] | P value | GRADE |
|---|---|---|---|---|---|---|
| Visual Analogue Scale[1] | | | | | | |
| | 6 weeks | 6 | 707 | 0.39 [-1.35, 2.13] | 0.66 | Low |
| | 3 months | 6 | 692 | -0.04 [-0.36, 0.29] | 0.83 | High |
| | 6 months | 7 | 722 | -0.06 [-0.30, 0.18] | 0.62 | High |
| | 1 year | 4 | 521 | -0.10 [-0.35, 0.15] | 0.45 | High |
| | 2 years | 4 | 551 | 0.11 [-0.12, 0.35] | 0.34 | High |

MD: mean difference, CI: confidence interval

[1]Scale from 0–10, a lower value is a better outcome.

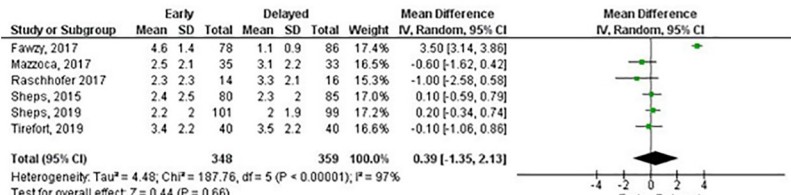

**Fig 3. Meta-analysis of pain at rest by Visual Analogue Scale at six weeks (a negative effect estimate favours early rehabilitation).**

## Outcomes

**Pain at rest.** Meta-analyses for pain at rest, measured with the Visual Analogue Scale, were possible at six weeks, three and six months, and one- and two-years follow-up. No statistically significant differences were found for any comparisons (Table 1). Figs 3 and 4 shows the forest plots at six weeks and three months, respectively. All forest plots for meta-analyses in Tables 1–4 that are not shown in the main text are available in the S5 File. The GRADE summary of findings with reasons for downgrading the certainty of effect is available in the S6 File.

**Function.** There was variation in the use of composite measures of shoulder pain and function and outcome measures for shoulder function. Given the wide range of data available, we combined studies according to the outcome measure used. Meta-analyses were possible for the ASES score (three and six months, and one and two years), Constant-Murley score (CM) (three and six months, and one and two years), Single Assessment Numeric Evaluation (SANE) (three and six months), Simple Shoulder Test (three and six months, and one year)

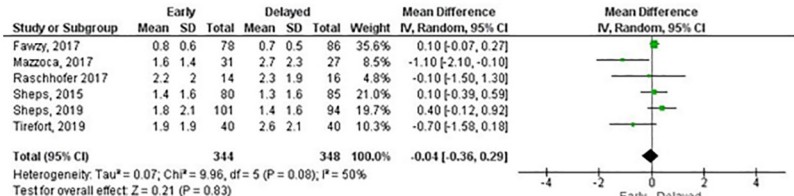

**Fig 4. Meta-analysis of pain at rest by Visual Analogue Scale at three months (a negative effect estimate favours early rehabilitation).**

**Table 2. Meta-analyses of function by outcome measures (A positive effect estimate favours early rehabilitation).**

| Outcome measure | Follow-up | Number of studies | Total sample size | Effect estimate MD [95% CI] | P value | GRADE |
|---|---|---|---|---|---|---|
| American Shoulder and Elbow score[1] | | | | | | |
| | 3 months | 3 | 243 | 3.43 [-1.07, 7.92] | 0.14 | Moderate |
| | 6 months | 4 | 365 | -0.26 [-4.76, 4.25] | 0.91 | Moderate |
| | 1 year | 4 | 441 | -0.57 [-2.39, 1.25] | 0.54 | Low |
| | 2 years | 2 | 202 | -2.67 [-6.35, 1.02] | 0.16 | Moderate |
| Constant-Murley score[1] | | | | | | |
| | 3 months | 4 | 313 | 3.18 [-1.53, 7.90] | 0.19 | Low |
| | 6 months | 6 | 513 | 1.19 [-1.33, 3.71] | 0.35 | High |
| | 1 year | 5 | 559 | -1.18 [-2.62, 0.25] | 0.11 | Low |
| | 2 years | 2 | 202 | -1.80 [-5.10, 1.49] | 0.28 | Moderate |
| Single Assessment Numeric Evaluation[1] | | | | | | |
| | 3 months | 2 | 138 | 2.23 [-4.62, 9.07] | 0.52 | Moderate |
| | 6 months | 2 | 138 | 6.54 [2.24, 10.84] | 0.003 | Moderate |
| Simple Shoulder Test[2] | | | | | | |
| | 3 months | 2 | 163 | -0.77 [-2.82, 1.28] | 0.46 | Very low |
| | 6 months | 3 | 277 | 0.63 [-0.36, 1.62] | 0.21 | Moderate |
| | 1 year | 3 | 277 | 0.39 [-0.40, 1.18] | 0.33 | Moderate |
| Western Ontario Rotator Cuff Index[1] | | | | | | |
| | 3 months | 2 | 309 | -1.82 [-5.96, 2.32] | 0.39 | Moderate |
| | 6 months | 2 | 305 | -1.29 [-5.17, 2.59] | 0.52 | Moderate |
| | 1 year | 2 | 300 | -1.91 [-5.21, 1.40] | 0.26 | Moderate |

MD: mean difference, CI: confidence interval

[1]Scale from 0–100, a higher value is a better outcome.

[2]Scale from 0–12, a higher value is a better outcome.

and Western Ontario Rotator Cuff Index (three and six months, and one year). There were no statistically significant differences for any outcome measures at any follow-up except for the SANE at six months in favour of early rehabilitation (Table 2). Overall, the mean differences were small in the short- and long-term follow-ups as illustrated in Figs 5 and 6 for the ASES. Other outcome measures used by RCTs included the Disabilities of the Arm, Shoulder and Hand, Oxford Shoulder Score, Shoulder Pain and Disability Index and University of California Los Angeles shoulder rating scale. The certainty of effects ranged from very low to moderate (Table 2).

**Range of movement.** Meta-analyses were possible for shoulder flexion, abduction, external rotation at 90° of abduction and internal rotation at 90° of abduction at six weeks, three and six months, and one and two years. All measurements were made using a goniometer. Statistically significant differences were found for flexion at six weeks, three and six months and one year, abduction at six weeks, external rotation at three and six months and internal rotation at six weeks, three and six months (Table 3). The certainty of effects ranged from very low to high.

**Strength.** Nine RCTs reported muscle strength [19–22, 24, 27–30]. Due to heterogeneity of testing position and data reporting, we did not pool the data into meta-analyses. Six RCTs used a hand-held dynamometer [19, 21, 22, 27, 28, 30]; other instruments included isokinetic dynamometer, tensiometer and the strength composite of the CM. The follow-up assessment ranged from six weeks to two years. Muscle strength was tested for flexion (five RCTs) [19, 21,

**Table 3. Meta-analyses of range of movement by movement (A positive effect estimate favours early rehabilitation).**

| Outcome measure (degrees) | Follow-up | Number of studies | Total sample size | Effect estimate MD [95% CI] | P value | GRADE |
|---|---|---|---|---|---|---|
| Flexion[1] | | | | | | |
| | 6 weeks | 6 | 753 | 7.36 [2.66, 12.06] | 0.002 | High |
| | 3 months | 10 | 1030 | 8.45 [3.43, 13.47] | 0.001 | Low |
| | 6 months | 12 | 1275 | 3.57 [0.81, 6.32] | 0.01 | Low |
| | 1 year | 9 | 1062 | 1.42 [0.21, 2.64] | 0.02 | Moderate |
| | 2 years | 4 | 543 | 1.61 [-1.11, 4.33] | 0.25 | High |
| Abduction[1] | | | | | | |
| | 6 weeks | 4 | 615 | 6.82 [2.30, 11.33] | 0.003 | Moderate |
| | 3 months | 5 | 581 | 6.68 [-1.47, 14.83] | 0.11 | Low |
| | 6 months | 5 | 574 | 1.14 [-2.78, 5.05] | 0.57 | Low |
| | 1 year | 4 | 529 | 0.05 [-4.04, 4.14] | 0.98 | Moderate |
| | 2 years | 2 | 341 | -1.73 [-7.41, 3.94] | 0.55 | Moderate |
| External rotation at 90° abduction[2] | | | | | | |
| | 6 weeks | 5 | 633 | 2.06 [-2.65, 6.76] | 0.39 | Moderate |
| | 3 months | 8 | 805 | 8.11 [3.85, 12.36] | <0.001 | Moderate |
| | 6 months | 9 | 964 | 1.77 [-0.05, 3.60] | 0.06 | Low |
| | 1 year | 7 | 839 | 0.76 [-2.01, 3.53] | 0.59 | Low |
| | 2 years | 3 | 461 | -0.70 [-5.69, 4.28] | 0.78 | Moderate |
| Internal rotation at 90° abduction[3] | | | | | | |
| | 6 weeks | 3 | 495 | 3.34 [0.40, 6.28] | 0.03 | High |
| | 3 months | 4 | 461 | 8.19 [0.99, 15.39] | 0.03 | Very low |
| | 6 months | 5 | 620 | 3.60 [0.06, 7.13] | 0.05 | Low |
| | 1 year | 4 | 580 | 1.26 [-1.64, 4.15] | 0.39 | Low |
| | 2 year | 2 | 341 | 0.54 [-2.39, 3.46] | 0.72 | Moderate |

[1] Scale from 0–180, a higher value is a better outcome.

[2] Scale from 0–90, a higher value is a better outcome.

[3] Scale from 0–70, a higher value is a better outcome.

22, 27, 28], abduction (four RCTs) [20–22, 27], internal (four RCTs) [19, 22, 28, 29] and external rotation (six RCTS) [19, 22, 27–30]. All RCTs reported that both groups improved in strength; another consistent finding across all RCTs was that no statistically significant differences between group were found for any strength test at any follow-up assessed.

**Repair integrity.** Meta-analyses were possible for three and six months and one-year follow-up. Only Arndt, Clavert [24] used a CT arthrography to assess the repair integrity, all other RCTs used ultrasound or MRI scan. There were no statistically significant differences between groups at any follow-up. At one year, the number needed to harm (NNH) was 42.5. We carried out a sensitivity analysis for the one-year meta-analysis by including only RCTs with overall low risk of bias. The sensitivity analysis showed a reduction in the odds ratio from 1.26 (95% CI: 0.82–1.93) (Fig 7) to 1.05 (95% CI: 0.64–1.75) (Fig 8); the NNH increased to 651. The certainty of effects ranged from low to moderate.

**Complications.** Overall, the number of complications were low. Nine RCTs [3, 20, 21, 23, 24, 27, 28, 31, 35] reported post-operative complications. The most common complication, reported in five RCTs [20, 21, 23, 24, 35], was related to limited shoulder range of movement.

**Table 4. Meta-analyses of repair integrity (re-tear events).**

| Outcome | Follow-up | Number of studies | Total sample size | Effect Estimate OR [95% CI] | P value | GRADE |
|---------|-----------|-------------------|-------------------|------------------------------|---------|-------|
| Repair integrity | | | | | | |
| | 3 months | 2 | 168 | 0.94 [0.39, 2.27] | 0.92 | Moderate |
| | 6 months | 3 | 221 | 1.34 [0.59, 3.04] | 0.48 | Moderate |
| | 1 year | 8 | 960 | 1.26 [0.82, 1.93] | 0.29 | Low |

OR: odds ratio, CI: confidence interval

Jenssen, Lundgreen [35] reported two cases of capsulitis in the early group and none in the delayed group. Sheps, Silveira [21] found one case of frozen shoulder in the early group and two cases in the delayed group. Koh, Lim [23] found no difference in the proportion of patients with stiffness (defined as having any one of the following three: forward elevation of <120˚, internal rotation lower than L3, and/or external rotation with the arm at the side of <20˚) at three months postoperatively (early: 53% vs delayed: 50%). However, at 2 years follow-up the proportion of patients with shoulder stiffness in the delayed group (38%, n = 18/48) was greater than in the early group (18%, n = 7/40). Arndt, Clavert [24] and Sheps, Bouliane [20] did not report the number of patients with complications by group. Other complications reported included deep and superficial infection, loose anchors, suture pull-out, persistent shoulder pain, biceps subluxation, detached biceps tendon, deep vein thrombosis and pulmonary embolism. No differences between groups were observed for these other complications.

**Patient and public involvement.** The only RCT to describe the participation of patients in the study development was Littlewood, Bateman [3]. Their patient and public involvement and engagement activities involved three meetings facilitated by the lead researcher. The group supported the co-production of patient-facing materials and development of study processes such as recruitment and informed consent. They also had a patient representative as part of their trial management group [36].

## Discussion

We aimed to summarise the effectiveness of early compared to delayed/standard rehabilitation following rotator cuff repair on clinical outcomes and repair integrity. We found no statistically significant differences for pain. For function, the only statistically significant difference was for the SANE at six months in favour of early rehabilitation (MD:6.54, 95%CI:2.24, 10.84, p = 0.003). The mean difference found for outcome measures for function were small and did not reach the minimal clinically important differences (MCID) [37–39]. The MCID for the

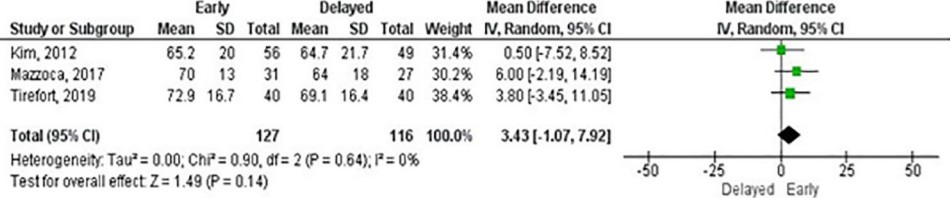

**Fig 5. Meta-analysis of function at 3 months by American Shoulder and Elbow Surgery score (A positive effect estimate favours early rehabilitation).**

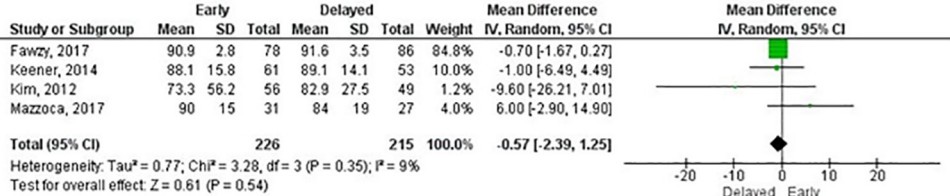

**Fig 6. Meta-analysis of function at 1 year by American Shoulder and Elbow Surgery score (A positive effect estimate favours early rehabilitation).**

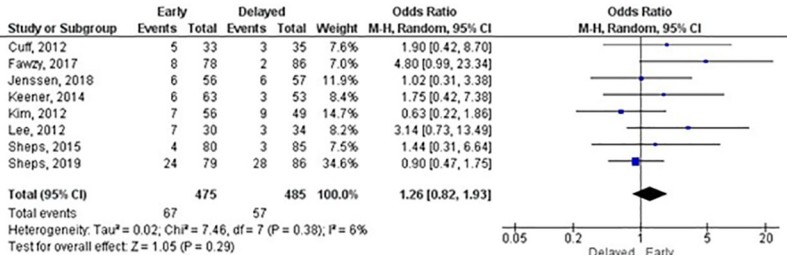

**Fig 7. Meta-analysis of repair integrity (re-tear events) at 1 year (An OR lower than 1 favours early rehabilitation).**

SANE for patients undergoing rotator cuff repair is reported to be 16.9 [38]. The meta-analyses for range of movement showed statistically significant differences for flexion, abduction, internal and external rotation, mainly in the short-term in favour of early rehabilitation. However, similar to function, the mean differences were small and unlikely to be clinically significant [40]. In terms of repair integrity, we also found no statistically significant differences between groups at any follow-up. Compared to the 2018 systematic review [8], we included another eight RCTs published since 2016 [3, 21, 26, 28, 30, 31, 34, 35], two RCTs [20, 29] that were excluded in the 2018 review due to the inclusion of patients with traumatic tears and the full text from Mazzocca, Arciero [33], which in the 2018 review was included only as an unpublished abstract. The addition of these other RCTs did not change the overall results and conclusions. The optimal approach to postoperative rehabilitation following surgical repair of the rotator cuff remain unclear. The majority of RCTs were rated at high risk or unclear for overall bias, this was mainly due to issues with the randomisation process, but also partially related to

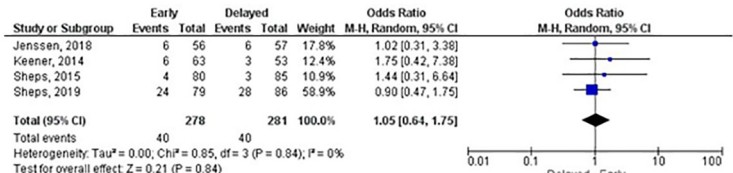

**Fig 8. Sensitivity meta-analysis of repair integrity (re-tear events) at 1 year (An OR lower than 1 favours early rehabilitation).**

the blinding of personnel and patients, which is not often possible in RCTs delivering exercises. Risk of bias was one of the main reasons for downgrading the certainty of the evidence.

Despite rotator cuff repair being such a common operation, we observed substantial variations in the rehabilitation protocols in relation to when patients were permitted to begin moving their shoulder and specific timelines for exercise progression. Almost all RCTs only allowed passive shoulder exercises in the first few weeks post-surgery for their early rehabilitation groups. Only Littlewood, Bateman [3], Sheps, Bouliane [20] used an individualised patient-directed approach, which facilitated a controlled and progressive introduction of active movements according to an acceptable symptoms response [41]. Sheps, Silveira [21] also used a patient-directed approach; however only pain-free active movements were allowed and exercise progression was still based on the number of weeks post-surgery. Restricting to only passive exercises in early postoperative stages may not provide sufficient load to stimulate and assist tendon healing [42] and, therefore, may not provide an optimal stimulus for tissue repair and remodelling to enable patients return to their usual activities, including leisure and work. We also found variations in the frequency of face-to-face appointments and duration of the programme.

Only four RCTs had clear distinctions between their groups regarding sling use; three RCTs prescribed the sling for comfort only and one RCT recommended patients to not use a sling at all. Although sling immobilisation has been traditionally viewed by many as important in protecting the tendon to facilitate healing, this is open to question. Stephens, Littlewood [43] interviewed patients who were part of the RaCeR RCT [3], where patients were supported to remove their sling after rotator cuff repair surgery as soon as they felt able. They reported that some patients who were in the delayed or standard rehabilitation group, and had to use a sling for four weeks, found that the sling contributed to their pain instead of relieving it. The restrictions imposed by the sling also impacted patients self-efficacy and in some cases even their self-identity. In contrast, some patients in the early rehabilitation group felt more confident and had the perception that moving their shoulder was contributing to their recovery. Another qualitative study [44] exploring patients perceptions of rehabilitation after shoulder arthroplasty reported that the sling, especially when using an abduction component, was impairing their sleep. Patients reported that they were unable to sleep because of the position, itching and temperature changes.

One of the main justifications for sling immobilisation and delayed rehabilitation following rotator cuff repair is the risk to the integrity of the repair, i.e. re-tear or failure to heal. However, as observed in our meta-analysis of repair integrity at one-year follow-up with RCTs at low risk of bias, the absolute risk reduction is very small (0.2%) and the NNH is 651. Thus, the chances of a re-tear that is caused by starting an early controlled and progressive mobilisation of the shoulder is low and no higher than delayed/standard rehabilitation; this needs to be considered in the context of the negative effects of immobilisation. We also undertook a sensitivity meta-analysis as four out of the eight RCTs included in the primary meta-analysis of repair integrity were rated at high risk of overall bias. Therefore, the sensitivity meta-analysis with only RCTs of low risk of bias provided a more reliable and robust result. There is debate that for small and medium tears early rehabilitation may be appropriate; however, for large tears the risk is less acceptable. Subgroups analyses by tear size was not possible and further recommendations for specific groups cannot be made. Only eight RCTs included patients with large tears in their sample, but data reported by tear size was not available. The RCT from Sheps, Silveira [21] (n = 206; low risk of bias), the largest RCT included in our systematic review, included patients with large tears. They found that despite patients with a large tear having a higher risk of a re-tear, this risk was not affected by the postoperative rehabilitation protocol. In contrast, using a sling for weeks may possibly cause problems such restricted range of

movement and may contribute to further deconditioning of an already weakened rotator cuff muscle. As observed by Koh, Lim [23], delaying mobilisation and restricting movement may increase the risk of patients having range of movement limitations in the long-term.

## Strengths and limitations

We followed strict methods for this systematic review. However, the certainty of effects were affected by the methodological quality of the body of evidence. The majority of the RCTs were considered of high or unclear overall risk of bias, had small sample sizes and their definition of early and delayed rehabilitation were not consistent. Further subgroup analyses were not possible due to the lack of data reported by tear size. Therefore, our results should be interpreted with caution [45].

## Implications for clinical practice

Current approaches to early rehabilitation following rotator cuff lead to earlier restoration of range of movement and, importantly, based on current data, risk of re-tear does not seem to be increased.

We found substantial variation in the time that patients used a sling, how exercises were progressed, and a lack of information about exercise dosage. This limits the ability to make specific clinical recommendations in relation to an optimal rehabilitation programme.

## Implications for research

Further large, high-quality RCTs are needed to investigate the effectiveness of early rehabilitation, particularly in relation to individual, patient-directed rehabilitation. Future RCTs must ensure that their sample size is of adequate power to allow to enable clinical recommendations with confidence.

## Conclusion

Although rotator cuff repair is a common surgery, postoperative rehabilitation has not evolved for over twenty years. The addition of new RCTs in this systematic review and meta-analysis has not changed the overall conclusion from a systematic review and meta-analysis reported in 2018. Rehabilitation protocols remain variable and cautious with regards to sling use and exercise progression. Our systematic review suggests that patients treated with early rehabilitation may regain range of movement earlier and are not at a higher risk of compromising the repair integrity, which has been a concern for clinicians. A large, high-quality multi-centre RCT, including all rotator cuff tear sizes and with a more progressive and individualised approach to early rehabilitation (gradual introduction of active use of the arm as soon as able and within acceptable limits of the individual patients' pain) is needed to advance knowledge and for conclusive recommendations on the optimal rehabilitation programme following rotator cuff repair.

## Supporting information

**S1 File. Search strategy.**
(DOCX)

**S2 File. Characteristics of the included RCTs.**
(DOCX)

**S3 File. Risk of bias by RCT.**
(DOCX)

**S4 File. Characteristics of the rehabilitation programmes.**
(DOCX)

**S5 File. Forest plots.**
(DOCX)

**S6 File. GRADE summary of findings.**
(DOCX)

**S1 Checklist.**
(DOC)

## Author Contributions

**Conceptualization:** Bruno Mazuquin, Chris Littlewood.

**Formal analysis:** Bruno Mazuquin.

**Methodology:** Bruno Mazuquin, Maria Moffatt, James Selfe, Chris Littlewood.

**Validation:** Maria Moffatt, Peter Gill.

**Writing – original draft:** Bruno Mazuquin, Chris Littlewood.

**Writing – review & editing:** Maria Moffatt, Peter Gill, James Selfe, Jonathan Rees, Steve Drew, Chris Littlewood.

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
