## [Decision Letter · Decision Letter 0]

29 Apr 2021

PONE-D-21-03503

Effectiveness of early versus delayed rehabilitation following rotator cuff repair: systematic review and meta-analyses

PLOS ONE

Dear Dr. Mazuquin,

Thank you for submitting your manuscript to PLOS ONE. After careful consideration, we feel that it has merit but does not fully meet PLOS ONE’s publication criteria as it currently stands. Therefore, we invite you to submit a revised version of the manuscript that addresses the points raised during the review process.

We look forward to receiving your revised manuscript.

Kind regards,

Katherine Saul

Academic Editor

PLOS ONE

Journal Requirements:

Reviewers' comments:

Reviewer's Responses to Questions

**Comments to the Author**

1. Is the manuscript technically sound, and do the data support the conclusions?

Reviewer #1: Yes

Reviewer #2: Yes

Reviewer #3: Yes

2. Has the statistical analysis been performed appropriately and rigorously? 

Reviewer #1: Yes

Reviewer #2: Yes

Reviewer #3: Yes

3. Have the authors made all data underlying the findings in their manuscript fully available?

Reviewer #1: Yes

Reviewer #2: Yes

Reviewer #3: Yes

4. Is the manuscript presented in an intelligible fashion and written in standard English?

Reviewer #1: Yes

Reviewer #2: Yes

Reviewer #3: Yes

5. Review Comments to the Author

Reviewer #1: General comments:

The authors present a review and meta-analysis of randomized controlled trials of rotator cuff tear patients with the objective to discern if early or delayed onset rehabilitation is most favorable. The motivation was to include 8 new studies in the analysis as perhaps new information could help shift the findings of prior reviews/meta-analyses. Ultimately the current results reflect the prior results. The paper is generally well written and easy to follow.

Specific comments:

Abstract

No comments, well written.

Introduction

Line 73: The overall motivation was simply to recreate the 2018 meta-analyses including 8 new studies. From the abstract it appears the same results are present with the addition of these 8 new studies. Should the review design have focused solely on the 8 new studies rather than include prior 12 studies , or at least teased out the new studies, so as to determine if a new effect is seen without influence of the prior results?

Methods

No comments, well written.

Results

Lines 288: Do the authors mean CT arthrography here?

Discussion

Lines 345: Conclude the first paragraph with the overall take home point, as this paragraph is mostly a high-level summary of the results. What is the primary message from this effort?

Line 349: “exercises”

The Discussion focuses heavily on the use of the sling. Can the authors also comment on the rehabilitation strategies in terms of the progression, and use of on-site versus home programs? This may speak to the cost-effectiveness argument raised on line 427.

Could the ultimately conclusion then be that addition of these 8 studies is more of the same…variable treatments of variable cuff tears in variable people don’t have a discernable effect on early versus delayed initiation of rehab protocols?

Reviewer #2: This systematic review and meta-analysis evaluated 20 randomized control trials (RCTs) to compare early rehabilitation versus delayed/standard rehabilitation after rotator cuff repair. Comparisons were made with regard to pain, function, range of motion, strength, and repair integrity. The need for this review exists because it remains unknown whether early or delayed/standard rehabilitation after rotator cuff repair has more favorable outcomes. A 2018 review on this topic included RCTs with high risk of bias and there was no data for large tears. Since 2018, 8 new RCTs were published, providing further rationale for an update on this topic. In the current review, only differences in range of movement were found in favor of early rehabilitation. Most clinical outcomes did not differ in early vs delayed rehab, and importantly there were no differences in repair integrity.

Overall, this is a nice review and meta-analysis. The following points of concern need to be addressed before this manuscript is suitable for publication.

The need for this review, as stated in the Intro, is that the 2018 meta-analysis included RCTs with high risk of bias and there was no data for large tears. However, it is unclear if the current review improved upon those limitations. In the current review, the majority of RCTS were considered of high risk or unclear risk, had small sample sizes, and unclear definitions of early and delayed rehab. Further, subgroup analyses weren't possible due to lack of data. Thus, it seems the limitations of the 2018 review were not improved upon. The authors should clearly discuss the findings of the current review in the context of the 2018 review. What same RCTs were included in both reviews? What new RCTs were included in the current review? How do the overall conclusions differ between the current review and the 2018 review?

Abstract - Line36-27. I don't understand what is meant by "number needed to harm was 651." This sentence is confusing.

Lines 139-140 Funnel plots weren't included per not recommended for meta-analyses with fewer than 10 RCTs. However, the current review included 20 RCTs so this is confusing.

Lines 149 Why weren't all 20 eligible RCTs included in the meta-analyses?

Line 196 - Visual Analog Scale should be all capitalized.

Figure captions should be improved. The Tables are great and easy to understand. However, the Figures need captions or subscripts to indicate the direction of the Mean Difference (early - delayed), or (delayed - early). The Odds Ratio should be similarly clarified.

Line 437 - review, not "reviewed"

Line 440 - Delete "that"

Line 442 - What is meant by "more progressive and controlled approach" ?

Reviewer #3: Rotator cuff pathologies are the most common shoulder pathology among the general population. The effective treatment of such pathologies is extremely important. The current manuscript reviewed the current state of the literature in order to assess the effectiveness of early vs. delayed rehabilitation following rotator cuff repair. I believe that this work is needed to inform clinical decision-making and improve the impact of current treatment paradigms for rotator cuff repairs in addition to other orthopaedic pathologies of the shoulder. This manuscript is well written and provides what appears to be a comprehensive survey of the literature. Below I provide line-by-line edits and general suggestions for each section of the manuscript.

General

Would it have been possible to also account for the repair and rehabilitation of individual rotator cuff muscle/tendon pathologies in addition to combining them into the rotator cuff?

Introduction

Lines 55-56: What proportion of people are managed surgically?

Lines 56-57: Is there a reason other than population growth why the number of procedures will increase?

Lines 68-69: Briefly let the reader know what is meant by repair integrity.

Line 74: Remove ‘an’.

Results

In general, I think it would be helpful to include units of measure in your tables/figures when possible.

Lines 290-291: Provide a short description of NNH and give some insight into what this number means.

Lines 290-295: It’s unclear what the reader should be taking from this information. It seems that by only including those with low risk for bias that the odds of suffering a retear decrease. Is that correct?

Lines 306-308 and 311-317: In these cases, what is the difference between limited range of motion and stiffness?

Conclusion

Line 437: Change ‘reviewed’ to ‘review’.

Line 440: Remove ‘that’.

6. PLOS authors have the option to publish the peer review history of their article (what does this mean?). If published, this will include your full peer review and any attached files.

Reviewer #1: No

Reviewer #2: No

Reviewer #3: No

---

## [Author Response · Author response to Decision Letter 0]

7 May 2021

Reviewer 1

General comments:

The authors present a review and meta-analysis of randomized controlled trials of rotator cuff tear patients with the objective to discern if early or delayed onset rehabilitation is most favorable. The motivation was to include 8 new studies in the analysis as perhaps new information could help shift the findings of prior reviews/meta-analyses. Ultimately the current results reflect the prior results. The paper is generally well written and easy to follow.

Specific comments:

Abstract

No comments, well written.

Thank you

Introduction

Line 73: The overall motivation was simply to recreate the 2018 meta-analyses including 8 new studies. From the abstract it appears the same results are present with the addition of these 8 new studies. Should the review design have focused solely on the 8 new studies rather than include prior 12 studies, or at least teased out the new studies, so as to determine if a new effect is seen without influence of the prior results?

The purpose of systematic reviews is to be comprehensive and include all relevant RCTs to minimise bias. Although a considerable number of RCTs have been published since the 2018 review, by excluding previous RCTs our review would risk publication bias. 

Methods

No comments, well written.

Thank you.

Results

Lines 288: Do the authors mean CT arthrography here?

Yes, the text has been amended accordingly.

Discussion

Lines 345: Conclude the first paragraph with the overall take home point, as this paragraph is mostly a high-level summary of the results. What is the primary message from this effort?

We have expanded the first paragraph to include the overall message of our paper. Lines 361-363.

‘The addition of these other RCTs did not change the overall results and conclusions. The optimal approach to postoperative rehabilitation following surgical repair of the rotator cuff remain unclear’

Line 349: “exercises”

The text has been amended

The Discussion focuses heavily on the use of the sling. Can the authors also comment on the rehabilitation strategies in terms of the progression, and use of on-site versus home programs? This may speak to the cost-effectiveness argument raised on line 427.

In keeping with usual practice, most programmes include a mix of clinic and home-based exercises and this is usually detailed in papers. We have discussed the findings of the different rehabilitation strategies from lines 369 to 384. 

‘Despite rotator cuff repair being such a common operation, we observed substantial variations in the rehabilitation protocols in relation to when patients were permitted to begin moving their shoulder and specific timelines for exercise progression. Almost all RCTs only allowed passive shoulder exercises in the first few weeks post-surgery for their early rehabilitation groups. Only Littlewood, Bateman (3), Sheps, Bouliane (20) used an individualised patient-directed approach, which facilitated a controlled and progressive introduction of active movements according to an acceptable symptoms response (41). Sheps, Silveira (21) also used a patient-directed approach; however only pain-free active movements were allowed and exercise progression was still based on the number of weeks post-surgery. Restricting to only passive exercises in early postoperative stages may not provide sufficient load to stimulate and assist tendon healing (42) and, therefore, may not provide an optimal stimulus for tissue repair and remodelling to enable patients return to their usual activities, including leisure and work. We also found variations in the frequency of face-to-face appointments and duration of the programme.’

Could the ultimately conclusion then be that addition of these 8 studies is more of the same…variable treatments of variable cuff tears in variable people don’t have a discernable effect on early versus delayed initiation of rehab protocols?

We have added the following sentence to the conclusion (line 457-459): ‘The addition of new RCTs in this systematic review and meta-analysis has not changed the overall conclusion from a systematic review and meta-analysis reported in 2018’

Reviewer 2

This systematic review and meta-analysis evaluated 20 randomized control trials (RCTs) to compare early rehabilitation versus delayed/standard rehabilitation after rotator cuff repair. Comparisons were made with regard to pain, function, range of motion, strength, and repair integrity. The need for this review exists because it remains unknown whether early or delayed/standard rehabilitation after rotator cuff repair has more favorable outcomes. A 2018 review on this topic included RCTs with high risk of bias and there was no data for large tears. Since 2018, 8 new RCTs were published, providing further rationale for an update on this topic. In the current review, only differences in range of movement were found in favor of early rehabilitation. Most clinical outcomes did not differ in early vs delayed rehab, and importantly there were no differences in repair integrity.

Overall, this is a nice review and meta-analysis. The following points of concern need to be addressed before this manuscript is suitable for publication.

The need for this review, as stated in the Intro, is that the 2018 meta-analysis included RCTs with high risk of bias and there was no data for large tears. However, it is unclear if the current review improved upon those limitations. In the current review, the majority of RCTS were considered of high risk or unclear risk, had small sample sizes, and unclear definitions of early and delayed rehab. Further, subgroup analyses weren't possible due to lack of data. Thus, it seems the limitations of the 2018 review were not improved upon. The authors should clearly discuss the findings of the current review in the context of the 2018 review. What same RCTs were included in both reviews? What new RCTs were included in the current review? How do the overall conclusions differ between the current review and the 2018 review?

We have now included in the discussion a paragraph detailing which RCTs were included in our new review and the impact that they had on the overall results and conclusions, Lines 356-363.

‘Compared to the 2018 systematic review (8), we included another eight RCTs published since 2016 (3, 21, 26, 28, 30, 31, 34, 35), two RCTs (20, 29) that were excluded in the 2018 review due to the inclusion of patients with traumatic tears and the full text from Mazzocca, Arciero (33), which in the 2018 review was included only as an unpublished abstract’

Abstract - Line36-27. I don't understand what is meant by "number needed to harm was 651." This sentence is confusing.

To make the information clearer, we changed the text to: ‘In terms of repair integrity, early mobilisation does not seem to increase the risk of re-tears (OR:1.05; 95%CI: 0.64-1.75)’.

Lines 139-140 Funnel plots weren't included per not recommended for meta-analyses with fewer than 10 RCTs. However, the current review included 20 RCTs so this is confusing.

Funnel plots are used to check for heterogeneity and publication bias in meta-analyses. Funnel plots are performed for individual meta-analyses. Given that none of our meta-analyses have more than 10 RCTs included, we have not created any Funnel plots. However, we have added the text (line 145)‘…, none of our meta-analyses had more than 10 RCTs’ 

Lines 149 Why weren't all 20 eligible RCTs included in the meta-analyses?

The meta-analyses combined data that was similar across RCTs. Each meta-analysis included RCTs that reported the same outcome. For example, not all RCTs reported shoulder flexion; therefore, only those reporting shoulder flexion were combined to undertake that meta-analysis. The same approach was followed for each meta-analysis. However, we have added the text in line 133-34 ‘The meta-analyses were presented according to the different outcome measures across RCTs’ to clarify this aspect of the analysis. 

Line 196 - Visual Analog Scale should be all capitalized.

Text has been amended

Figure captions should be improved. The Tables are great and easy to understand. However, the Figures need captions or subscripts to indicate the direction of the Mean Difference (early - delayed), or (delayed - early). The Odds Ratio should be similarly clarified.

We have amended figures’ captions as requested by adding information about the direction of the results.

Line 437 - review, not "reviewed"

Text has been amended

Line 440 - Delete "that"

Text has been amended

Line 442 - What is meant by "more progressive and controlled approach"?

We have added more information to clarify what we mean by a more progressive and controlled approach, line 465-467: ‘gradual introduction of active use of the arm as soon as able and within acceptable limits of pain’.

 

Reviewer 3

Rotator cuff pathologies are the most common shoulder pathology among the general population. The effective treatment of such pathologies is extremely important. The current manuscript reviewed the current state of the literature in order to assess the effectiveness of early vs. delayed rehabilitation following rotator cuff repair. I believe that this work is needed to inform clinical decision-making and improve the impact of current treatment paradigms for rotator cuff repairs in addition to other orthopaedic pathologies of the shoulder. This manuscript is well written and provides what appears to be a comprehensive survey of the literature. Below I provide line-by-line edits and general suggestions for each section of the manuscript.

General

Would it have been possible to also account for the repair and rehabilitation of individual rotator cuff muscle/tendon pathologies in addition to combining them into the rotator cuff?

In our PROSPERO protocol, we defined that subgroup analyses, if sufficient data were available, would be undertaken for type of repair method (single or double-row), age group (<65 or >65 years old), tear sizes (small, medium, large and massive) and tear type (traumatic or non-traumatic). However, due to the lack of data reported in RCTs, these sub-group analyses were not possible. We did not pre-specify a subgroup analysis by tendons affected. Although we agree that this could be an interesting analysis, we were aware that RCTs rarely, if ever, report this data. 

Introduction

Lines 55-56: What proportion of people are managed surgically?

The proportion of patients who are managed surgically is difficult to estimate. According to Kuhn et al. (2013), approximately 75% of patients who are treated first with physiotherapy, do not subsequently undergo surgery. However, we have included a sentence and a new reference indicating the number of surgeries undertaken annually in the UK; line 59: ‘In 2018/2019, almost 9,000 surgical repairs were undertaken in the UK National Health Service’ (4)

Kuhn, J.E., Dunn, W.R., Sanders R. et al. Effectiveness of physical therapy in treating atraumatic full-thickness rotator cuff tears: a multicenter prospective cohort study. J Shoulder Elbow Surg. 2013 Oct;22(10):1371-9. doi: 10.1016/j.jse.2013.01.026.

Lines 56-57: Is there a reason other than population growth why the number of procedures will increase?

There are a couple of factors that have been attributed to the increase on the number of rotator cuff repairs. According to Palovena et al, (2015), potential factors associated with this increase are related to the growing awareness of rotator cuff disorders, the increased availability of diagnostic (i.e. radiological imaging) methods and arthroscopic surgery, and advancements in surgical techniques. However, population growth is likely to be the main factor.

Paloneva, J., Lepola, V., Äärimaa, V. et al. Increasing incidence of rotator cuff repairs—A nationwide registry study in Finland. BMC Musculoskelet Disord 16, 189 (2015). https://doi.org/10.1186/s12891-015-0639-6

Lines 68-69: Briefly let the reader know what is meant by repair integrity.

We have added the sentence ‘i.e. the number of re-tear events was similar between groups’ to line 71.

Line 74: Remove ‘an’.

Text has been amended

Results

In general, I think it would be helpful to include units of measure in your tables/figures when possible.

We have added the unit of measure to Table 3 – range of movement. All other outcomes are unitless.

Lines 290-291: Provide a short description of NNH and give some insight into what this number means.

We have provided an explanation about the NNH in the discussion, lines 406-409: ‘Thus, the chances of a re-tear that is caused by starting an early controlled and progressive mobilisation of the shoulder is low and no higher than delayed/standard rehabilitation; this needs to be considered in the context of the negative effects of immobilisation’

Lines 290-295: It’s unclear what the reader should be taking from this information. It seems that by only including those with low risk for bias that the odds of suffering a retear decrease. Is that correct?

Yes, that’s correct. In this sensitivity analysis we wanted to show the results based only on RCTs at low risk of bias. By completing this analysis, we present a more reliable and robust result, and show that RCTs of lower methodological quality are affecting the effect estimate. We have expanded the discussion to include the rationale for the sensitivity meta-analysis, lines 409-413: ‘We also undertook a sensitivity meta-analysis as four out of the eight RCTs included in the primary meta-analysis of repair integrity were rated at high risk of overall bias. Therefore, the sensitivity meta-analysis with only RCTs of low risk of bias provided a more reliable and robust result.’

Lines 306-308 and 311-317: In these cases, what is the difference between limited range of motion and stiffness?

In this context both terms could be used interchangeably. However, we have changed the word ‘stiffness’ for ‘restricted range of movement’ for consistency, line 423.

Conclusion

Line 437: Change ‘reviewed’ to ‘review’.

Text has been amended

Line 440: Remove ‘that’.

Text has been amended

---

## [Editor Report · Decision Letter 1]

11 May 2021

Effectiveness of early versus delayed rehabilitation following rotator cuff repair: systematic review and meta-analyses

PONE-D-21-03503R1

Dear Dr. Mazuquin,

We’re pleased to inform you that your manuscript has been judged scientifically suitable for publication and will be formally accepted for publication once it meets all outstanding technical requirements.

Kind regards,

Katherine Saul

Academic Editor

PLOS ONE
---

## [Editor Report · Acceptance letter]

19 May 2021

PONE-D-21-03503R1 

Effectiveness of early versus delayed rehabilitation following rotator cuff repair: systematic review and meta-analyses 

Dear Dr. Mazuquin:

I'm pleased to inform you that your manuscript has been deemed suitable for publication in PLOS ONE. Congratulations! Your manuscript is now with our production department. 

Kind regards, 

on behalf of

Dr. Katherine Saul 

Academic Editor

PLOS ONE